# Mechanical Properties of 3D-Printed Occlusal Splint Materials

**DOI:** 10.3390/dj11080199

**Published:** 2023-08-18

**Authors:** Vladimir Prpic, Filipa Spehar, Dominik Stajdohar, Roko Bjelica, Samir Cimic, Matej Par

**Affiliations:** 1Department of Fixed Prosthodontics, School of Dental Medicine, University of Zagreb, 10000 Zagreb, Croatia; vprpic@sfzg.hr; 2School of Dental Medicine, University of Zagreb, 10000 Zagreb, Croatia; fspehar@sfzg.hr (F.S.); dstajdohar@sfzg.hr (D.S.); 3Department of Oral Surgery, School of Dental Medicine, University of Zagreb, 10000 Zagreb, Croatia; rbjelica@sfzg.hr; 4Department of Removable Prosthodontics, School of Dental Medicine, University of Zagreb, 10000 Zagreb, Croatia; 5Department of Endodontics and Restorative Dentistry, School of Dental Medicine, University of Zagreb, 10000 Zagreb, Croatia; mpar@sfzg.hr

**Keywords:** 3D-printed, milled, flexural strength, surface hardness, occlusal splint

## Abstract

Data regarding the mechanical properties of three-dimensionally (3D) printed materials for occlusal splint manufacturing are scarce. The aim of the present study was to evaluate the flexural strength and surface hardness of modern 3D-printed occlusal splint materials and compare them with two control groups, namely, milled and conventional cold-polymerized occlusal splint materials. A total of 140 rectangular specimens were manufactured for the present study. The specimens were prepared in accordance with the International Organization for Standardization standards (ISO 20795-1:2013). Five 3D-printed (NextDent Ortho Rigid, Dental LT Clear, Dentona Flexisplint, Cosmos Bite Splint, and ProArt Print Splint), one milled (ProArt CAD Splint), and one cold-polymerized (ProBase Cold) occlusal splint materials were used to determine flexural strength and surface hardness values. The three-point flexure test was used for the determination of flexural strength values, while Vickers hardness was measured to determine surface hardness. Ten specimens (*n* = 10) of each material were tested using these procedures. One-way ANOVA and Tukey’s post-hoc test were used to analyze the obtained results (α = 0.05). The values of flexural strength ranged from 46.1 ± 8.2 MPa to 106 ± 8.3 MPa. The Vickers hardness values ranged from 4.9 ± 0.5 VHN to 20.6 ± 1.3 VHN. Significant differences were found among the tested materials (*p* < 0.0001). The milled and cold-polymerized materials yielded higher values for both flexural strength (only one 3D-printed resin had comparable results to cold-polymerized acrylics) and surface hardness. There are differences in the mechanical properties of the various tested occlusal splint materials. The flexural strength of most of the 3D-printed materials and their surface hardness values are still inferior when compared to the milled or cold-polymerized materials.

## 1. Introduction

Occlusal splints represent the main therapeutic option for patients suffering from temporomandibular disorders (TMDs) and bruxism [1]. They are removable apparatuses positioned on the upper or lower jaw that affect the fitting between the jaws. Their implementation has been proven to be successful in 70–90% of TMD cases [2]. Also, occlusal splints offer benefits in relation to stopping or reducing tooth wear among patients suffering from bruxism.

Since high occlusal forces can be generated in patients’ mouths (in bruxers values ranging from 450 N to 650 N, with a mean value of 380 N) [2], the materials used for the manufacturing of occlusal splints must have optimal mechanical properties [3,4]. The material of choice for the manufacturing of occlusal splints is still poly(methyl methacrylate) (PMMA) [5]. Currently, several workflows for the fabrication of occlusal splints are used. In the conventional workflow, occlusal splints are made of cold-polymerized PMMA in a powder-liquid form, whose optimal mechanical properties gave rise to its gold-standard status for the treatment of TMDs and bruxism [3,5,6,7]. Poly(methyl methacrylate) has benefits including esthetics and optimal chemical and physical qualities while also being relatively inexpensive [8]. Nevertheless, a great quantity of residual monomer and polymerization shrinkage can influence the mechanical properties of occlusal splints and decrease their clinical performance [6,9]. The incomplete conversion of (co-)monomers to polymers, even under optimum polymerization conditions, is a disadvantage of all materials based on methacrylates. The production process variables such as the monomer-to-polymer ratio and polymerization time and temperature have a major impact on the degree of conversion. Free residual (co-)monomers, additives, and reaction products can potentially be released as a result of insufficient monomer conversion [8], which weakens the mechanical properties of the (cold-polymerized) resin.

Recently, digitally supported fabrication processes (computer-aided design/computer-aided manufacturing (CAD/CAM)) have allowed for the usage of additive (3D-printed) and subtractive (milling) technologies and given rise to an improvement in dental medicine [5,6,10,11,12]. In subtractive methods, occlusal splints are milled from a prefabricated poly(methyl methacrylate) disc [3,9,13]. Subtractive manufacturing is not novel; it appeared in dental medicine in the late 1980s. Since then, milling materials have undergone significant improvements. Studies have shown that they possess better physical properties when compared with materials used in conventional processing methods [8]. The milled objects (e.g., occlusal splints) fit better because of reduced polymerization shrinkage of the industrially produced PMMA discs, which have a high double-bond conversion rate. The ability to produce occlusal devices more quickly and consistently than via manual fabrication is a benefit of the milling technique [14].

Contrary to milling manufacturing, additive manufacturing, also known as three-dimensional (3D) printing, was first proposed in 1986, and it took some time for the (dental) industry to accept it [14,15]. Reduced material waste is one of the major advantages of additive manufacturing over milling with respect to obtaining certain objects [14]. Other advantages include the ability to create complex geometries, reproducibility, simple use and ease of production, high productivity, and cost-effectiveness. Stereolithography (SLA) and digital light processing (DLP) are two types of additive-manufacturing processes, and they are currently most widely used when fabricating occlusal splints [3,4,5,16]. In SLA, photopolymers change from liquid to solid form below ultraviolet (UV) light, whilst in DLP, photopolymers are exposed to light released from a projector [17]. The main difference between them is the source of light, to whom the photopolymers are exposed until the definitive object (an occlusal splint) is formed layer upon layer [17]. Most frequently, the photopolymer resin is made of 75% oligomers, 25% monomers, and photopolymerization initiators. The initiators are primary radical cleavers that, when exposed to ultraviolet light, react with the oligomers and monomers to form polymers [18,19]. The production of occlusal splints is now possible thanks to the recent advancement of durable and biocompatible polymers for 3D printers. Still, it should be noted that 3D printing is a relatively new technique in dental medicine that is still evolving. Studies have shown that the choice of the material, the printing resolution, the post-curing conditions, speed, and the intensity of the laser have an impact on the accuracy of 3D-printed dental devices [20].

For the production of occlusal splints, digital technologies must deliver predictable, ideal, and consistent results [21]. These resins are mostly methylmethacrylate-based and share many characteristics with traditional dental acrylic resin. Regulatory organizations in the United States and Europe have recently approved the usage of these products in vivo [21]. Vasques et al. [21] and Lauren et al. [22] state that the design and manufacturing of occlusal splints by using digital scanning and CAD/CAM technology have several benefits, including reproducibility, an improvement in the internal fit of a device on a dental arch and the accuracy of the occlusal contacts, better resin polymerization, a lack of distortion or contractions, and less time required for appliance fabrication and adjustment. Due to their expensiveness and dentists’ lack of experience in using them, 3D printers are difficult to purchase, which Brandt et al. [23] and Velasco et al. [24] suggest is the biggest drawback of digital technology in dental medicine. Along with the already-mentioned minimization of material waste, another benefit of 3D-printing technology is that it can create multiple objects at one time. It can also be used to create complex designs involving multiple undercuts [18,19,25]. Moreover, 3D printing is an efficient, profitable, and predictable fabrication method [10,26] with the capacity for further development. Still, the materials used for the additive manufacturing of occlusal splints differ from the conventional and subtractive materials. Recent available studies [3,5,27] have demonstrated that the properties (e.g., flexural strength, surface hardness, and wear resistance) of additive-manufacturing materials for occlusal splint fabrication are generally inferior when compared to conventional PMMA or milled materials, which is the reason they are still less used than conventional and milled materials for the fabrication of occlusal splints. Still, it must be noted that the mechanical properties of occlusal splints depend to a greater degree upon the material composition rather than the manufacturing technology [5]. Further frequent applications of 3D printing (photopolymers) in prosthodontics include provisional restorations, implant drilling guides, complete dentures, and diagnostic and adjuvant appliances [28,29,30]. Even though past studies have shown the generally inferior mechanical properties of their products, 3D printing represents a promising scientific and technological field through which great development and enhanced materials are attainable.

The present in vitro investigation aimed to evaluate the mechanical properties (flexural strength and surface hardness) of various 3D-printed materials used for occlusal splint manufacturing currently available on the market and compare them to control groups (milled and cold-polymerized). The research hypotheses were that the flexural strength and surface hardness values would be comparable among the various materials tested.

## 2. Materials and Methods

Five 3D-printed, one cold-polymerized (first control group), and one milled material (second control group) for occlusal splint manufacturing were selected in this in vitro study. Table 1 shows a list of the used materials, manufacturers, compositions, and occlusal-splint-manufacturing methods. These are all materials available on the market, and some of them differ in their chemical compositions. Two of the examined materials (ProArt CAD Splint (PCS) and ProBase Cold (PBC)) were poly(methyl methacrylate) in terms of composition. Four materials (Dental LT Clear (DLC), Dentona Flexisplint (DFS), Cosmos Bite Splint (CBS), and ProArt Print Splint (PPS)) were acrylic light-polymerizing resins, whilst one of the materials (NextDent Ortho Rigid (NOR)) has been declared to be monomer based on acrylic esters. A total of 140 specimens with precise dimensions of 64.0 × 10.0 × 3.3 ± 0.2 mm (Figure 1) were manufactured for the flexural strength and surface hardness measurements. Each examined group contained 10 specimens (*n* = 10) for both flexural strength and surface hardness testing. The relevant International Organization for Standardization (ISO) standard (ISO 20795-1:2013) [31] was followed when conducting the flexural strength tests.

To obtain specimens made from a conventional material, a block of cold-polymerized occlusal splint material (ProBase Cold) was prepared from a condensation (C)-silicone (Optosil^®^, Kulzer GmbH, Hanau, Germany) mold. Previously, a rectangular block (the shape of the future acrylic block) of dental gypsum was made and immersed in C-silicone, which was mixed according to the manufacturer’s recommendations. Afterwards, monomer and polymer components (cold-polymerized materials) were mixed in adequate proportions and poured into the C-silicone mold. The C-silicone mold, which was filled with cold-polymerized material (ProBase Cold), was placed into a pressure-polymerizing apparatus (Ivomat IP2, Ivoclar Vivadent AG, Schaan, Liechtenstein) with 6 bar of pressure at 40 °C for 15 min.

For the 3D-printed samples, two different software products (Netfabb Premium 2019, Autodesk, San Rafael, CA, USA; Meshmixer, Autodesk, San Rafael, CA, USA) were used to create and prepare virtual rectangular blocks in standard tessellation language (STL) form prior to additive manufacturing. Rectangular blocks of 3D-printed occlusal splint materials (NextDent Ortho Rigid, Dental LT Clear, Dentona Flexisplint, Cosmos Bite Splint, and ProArt Print Splint) were prepared in accordance with the obtained STL file in a suitable 3D printer with a subsequent post-polymerization procedure. Both 3D-printing and the post-polymerization procedure were conducted according to the manufacturer’s recommendations. Cosmos Bite Splint was printed using P20+ (Straumann, Basel, Switzerland) 3D printer with a subsequent post-polymerization procedure (P-Cure, Straumann, Basel, Switzerland); Dental LT Clear was printed using Form 3B (Formlabs, Somerville, MA, USA), followed by a post-polymerization procedure in Form Cure (Formlabs, Somerville, MA, USA); NextDent Ortho Rigid was printed using AccuFab-L4D (Shining 3D GmbH, Hangzhou, China), followed by post-polymerization procedure in FabCure (Shining 3D GmbH, Hangzhou, China); ProArt Print Splint was printed using PrograPrint PR5 (Ivoclar Vivadent AG, Schaan, Liechtenstein), followed by post-polymerization procedure in PrograPrint Cure (Ivoclar Vivadent AG, Schaan, Liechtenstein); and Dentona Flexisplint was printed using LC Opus (Photocentric Ltd., Peterborough, UK), followed by post-polymerization procedure in Cure M+ (Photocentric Ltd., Peterborough, UK).

Since milled materials come prefabricated as a single pre-polymerized block, the used milled material (ProArt CAD Splint) was not treated before the cutting of the samples (out of blocks).

Specimens with approximate dimensions to the final ones (64.0 × 10.0 × 3.3 ± 0.2 mm) (Figure 1) were cut from the prepared cold-polymerized (ProBase Cold) block, milled (ProArt CAD Splint) a prefabricated block, and the obtained 3D-printed blocks (NextDent Ortho Rigid, Dental LT Clear, Dentona Flexisplint, Cosmos Bite Splint, and ProArt Print Splint) on a water-cooled cutting engine (IsoMet 1000, Buehler, Lake Bluff, IL, USA) (so that the generated temperatures did not exceed 30 °C during the shaping of the specimens).

In addition, all surfaces were further reduced by employing standard metallographic grinding papers with grain sizes of approximately 30 μm (P500), 18 μm (P1000), and 15 μm (P1200) on a polishing machine (Le Cube, Presi, Eybens, France) to obtain definitive dimensions with smooth surfaces. The three-point bending test was used for the determination of flexural strength in accordance with the cited ISO standard (ISO 20795-1:2013) [31]. The flexural strength of the specimens was measured using a universal testing instrument (Inspekt Duo 5 kN-M, Hegewald & Peschke GmbH, Nossen, Germany). According to the given ISO standard [31], the specimens were kept in a water bath for 50 ± 1 h at 37 °C prior to the testing process. After the recommended time had passed, each specimen was carefully removed from the water bath and placed on the cylindrical carriers of the testing instrument. The distance between the centers of the carriers was 50 mm, and the loading penetrant was positioned halfway between the supporting carriers. The load that was applied to the specimens was amplified from 0 by using a crosshead speed of 1 mm/min until the specimen broke. Flexural strength was computed in accordance with the following equation: FS = 3FL/2bh^2^, where FS stands for flexural strength (MPa), F stands for maximal load (N), L stands for the spacing between the carriers (mm), b stands for the wideness of the specimen (mm), and h stands for the height of the specimen (mm).

Surface hardness was calculated using the Vickers hardness test using the following formula: HV = 1.854(F/D^2^), where HV stands for Vickers hardness (VHN), F stands for load applied to a specimen (kgf), and D stands for area of the indentation (mm^2^). Hardness tester (CSV-10, ESI Prüftechnik GmbH, Wendlingen am Neckar, Germany) was used for the determination of Vickers hardness. A 100 g load was applied to each specimen’s surface for a dwelling time set at 10 s. Five measurements were taken for each specimen, and, eventually, the mean Vickers hardness value was calculated and further used in the statistical analysis.

Shapiro–Wilk test was used to evaluate the normality of data, and Levene’s test was used to assess the equality of variances. Descriptive statistics were computed (mean, SD, and minimal and maximal values). One-way ANOVA and Tukey post-hoc tests were conducted to collate all examined groups. The analysis was carried out on the Windows platform (Microsoft Corporation, Redmond, WA, USA) with the statistical package SAS (SAS 8.2, SAS, Cary, NC, USA). All tests were run at a significance level of α = 0.05.

## 3. Results

The mean values and standard deviations for the obtained flexural strength and Vickers hardness values are depicted in Figure 2 and Figure 3. In the course of flexural strength testing, 10 specimens of ProArt Print Splint and 2 specimens of Dentona Flexisplint did not break during the loading within the extreme limits of the loading plunger’s possible movements. Accordingly, the flexural strength values of ProArt Print Splint could not be measured and were not included in the statistical analysis. The flexural strength values of Dentona Flexisplint were measured for eight specimens; accordingly, statistical analysis of the mentioned material was only performed for eight specimens (instead of ten specimens like in the other groups). The maximum and minimum values of flexural strength (MPa) of the examined materials were as follows: 122.1 and 90.2 for ProArt CAD Splint, 87.8 and 77.4 for ProBase Cold, 82.9 and 57.6 for Dental LT Clear, 87.9 and 59.3 for NextDent Ortho Rigid, 50.8 and 40.7 for Dentona Flexisplint, and 57.7 and 33.5 for Cosmos Bite Splint. The maximum and minimum values of Vickers hardness (VHN) of the examined groups were 23.1 and 19.1 for ProArt CAD Splint, 20.7 and 17.4 for ProBase Cold, 14.3 and 8.9 for NextDent Ortho Rigid, 13.2 and 9.5 for Dental LT Clear, 9.6 and 8.1 for Dentona Flexisplint, 11.3 and 6.6 for Cosmos Bite Splint, and 5.8 and 4.2 for ProArt Print Splint. Two 3D-printed materials (Dentona Flexisplint and Cosmos Bite Splint) did not meet the ISO requirements [31] for flexural strength (≥65 MPa). One-way ANOVA showed statistically significant (*p* < 0.0001) differences of flexural strength and Vickers hardness between the examined groups. The results of the Tukey post-hoc test are shown in Figure 2 and Figure 3. The flexural strength values of ProArt CAD Splint were the highest and were statistically different from those of all the other groups (*p* < 0.05). The results regarding ProBase Cold and Dental LT Clear did not statistically differ (*p* ≥ 0.05) from each other, nor did those regarding NextDent Ortho Rigid and Dental LT Clear (*p* ≥ 0.05) or Dentona FlexiSplint compared to Cosmos Bite Splint (*p* ≥ 0.05). The Vickers hardness values (VHN) of ProArt CAD Splint, ProBase Cold, and ProArt Print Splint statistically differed from those of all the tested materials and from each other (*p* < 0.05). The VHN between NextDent OrthoRigid and Dental LT Clear did not statistically differ (*p* ≥ 0.05). Similarly, the VHN between Dentona Flexisplint and Cosmos Bite Splint statistically differed from all the other groups but not from each other (*p* ≥ 0.05).

## 4. Discussion

This study examined various additively manufactured occlusal splint materials with respect to their flexural strength and surface hardness properties and compared them with two control groups, using cold-polymerized and milled acrylics as base materials. Generally, the 3D-printed materials did not demonstrate mechanical properties as high as the milled or cold-polymerized materials. Therefore, the research hypothesis that the flexural strength and surface hardness values would be comparable among various materials was rejected.

Flexibility of dental materials is crucial for energy absorption in instances where patients drop an oral apparatus [32]. Accordingly, a minimal flexural strength value of 65 MPa is specified for optimal performance by the ISO 20795-1:2013 standard [31]. Therefore, the cited ISO standard was used to carry out flexural strength testing. In the present in vitro study, milled PMMA material (ProArt CAD Splint) had the highest value of flexural strength (Figure 2). The 3D-printed material with the highest values of flexural strength was Dental LT Clear (75.2 ± 7.6 MPa), whose values did not statistically differ from those of the cold-polymerized acrylics (ProBase Cold; (82.2 ± 3.9 MPa)). This result is comparable to that obtained in the investigation by Berli et al. [3], where one (Nextdent Ortho Clear) out of three additive-manufacturing groups also had similar flexural strength values to cold-polymerized acrylic resins (ProBase Cold and Aesthetic Blue clear). In view of these findings, 3D-printed materials for occlusal splint fabrication could exhibit similar flexural strength values to conventional cold-polymerized materials with further development. Still, at this point, the 3D-printed materials available on the market generally have lower flexural strength values in comparison with milled and cold-polymerized acrylics. The opposite results to those of the Dental LT Clear group were displayed by two 3D-printed materials (Dentona Flexisplint and Cosmos Bite Splint, as shown in Figure 2), whose values did not meet the ISO recommendations for flexural strength. For the fifth group (ProArt Print Splint), flexural strength values could not be measured since neither specimen broke during flexural strength testing. This phenomenon (samples not breaking) can be found in the literature [5,32], and it can be considered normal for these types of tests when applied to such specimens. In the study by Berli et al. [3], the authors analyzed three 3D-printed, three milled, and three pressed occlusal splint materials and reported that the milled materials yielded higher flexural strength values than the 3D-printed ones, constituting a result that is comparable with this investigation’s results (Figure 2). Interestingly, in both studies, i.e., Berli et al.’s [3] and the present one, (Figure 2) two 3D-printed materials did not meet the ISO standard requirements [31] for flexural strength. The insufficient values in the present study ranged from 46.1 ± 8.2 MPa for Cosmos Bite Splint to 46.3 ± 3.3 MPa for Dentona Flexisplint (Figure 2), while in the study by Berli et al. [3], the values ranged from 19.5 ± 2.5 MPa for Freeprint splint to 39.3 ± 2.0 MPa for LuxaPrint Ortho Plus. Accordingly, it is questionable whether the given standard is adequate for the testing of materials for additive manufacturing (3D printing). Also, based on the findings of the present study (Figure 2) and those in the study by Berli et al. [3], it can be concluded that these materials should be used with caution in everyday clinical practice.

Generally, the surface hardness of a material is an indicator of its resistance to wear, and it shows whether a certain material is capable of resisting scratches and possible dimensional alterations [33,34]. The high occlusal forces that can develop (especially among bruxers, but also among other patients) demand high surface hardness values of the used occlusal splint materials [5]. In the present study, the Vickers hardness values of the additive-manufacturing occlusal splint materials were significantly lower in comparison to those of the milled or cold-polymerized materials. The 3D-printed materials with the highest Vickers hardness values were NextDent Ortho Rigid, with 11.3 ± 1.7 VHN, and Dental LT Clear, with 11.2 ± 1.0 VHN (Figure 3). In the study by Berli et al. [3], the hardness values (Martens and Vickers hardness) of 3D-printed materials could not be measured because these materials demonstrated hardness values under the testing standard. Two additive manufacturing materials in the study by Prpic et al. [5] had lower values of surface hardness than most of the other tested materials (milled and cold-polymerized materials). Similar findings were recently published by Gibreel et al. [2], where the authors compared the hardness of milled and additively manufactured materials in relation to the conventional manufacturing of occlusal splints. To conclude, 3D-printed materials have lower surface hardness values in comparison with conventional materials [2,3,5] (Figure 3). These results could be clarified by the layer deposition, which was parallel to the loading orientation in the 3D-printing techniques, leading to low mechanical properties, as well as the fact that the strength of adhesion between subsequent layers is less than the strength of each individual layer [2]. Differences between different materials and fabrication techniques should be taken into account when choosing the optimal solution for the treatment of patients. Based on the findings of the present study (Figure 3) and previous studies [2,5] on hardness, 3D-printed occlusal splints cannot be recommended for use in long-term therapy.

Additive technologies (3D printing) applied for occlusal splint fabrication could be beneficial in terms of reducing chair time, possible mistakes, and costs [3]. Based on the present study’s findings, milled materials for occlusal splint fabrication showed superior mechanical properties (Figure 2 and Figure 3) and can be highlighted as optimal materials for the manufacturing process. Nevertheless, present and future developments in the field of additive manufacturing could bring novel 3D-printed materials with mechanical properties that could compete with those of other materials and processing techniques; however, for now, modern 3D-printed materials are generally inferior (Figure 2 and Figure 3) [2,3,5]. However, 3D-printed materials could be used in short-term therapy. It would be advantageous if future studies could focus on the clinical performance of 3D-printed occlusal splints and their longevity (in the short and long term) and efficacy with respect to the oral environment (in vivo studies) as well as on the precision of certain types of 3D printers and post-polymerization processes. The limitation of the present investigation is that the effects of the printing direction and the layer thickness of 3D-printed materials were not analyzed, which could have influenced the outcome of the study.

## 5. Conclusions

The findings of the present in vitro investigation showed that milled occlusal splint materials exhibit the highest mechanical properties (i.e., flexural strength and surface hardness) in comparison to cold-polymerized or additively manufactured occlusal splint materials. Even though the cold-polymerized material (ProBase Cold) had higher flexural strength values when compared to one 3D-printed material (Dental LT Clear), statistically significant differences between these two materials were not perceived (Figure 2). This could indicate that the improvement of additive technologies and 3D-printed materials might guarantee their optimal mechanical properties, which could expand their utilization in the field of prosthodontics. But today’s 3D-printed materials generally still have lower mechanical properties compared to cold-polymerized acrylics.

## Figures and Tables

**Figure 1 dentistry-11-00199-f001:**
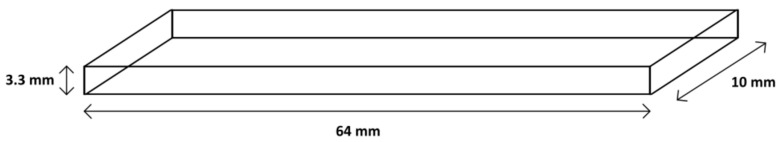
Scheme (dimensions) of specimens tested in the present study.

**Figure 2 dentistry-11-00199-f002:**
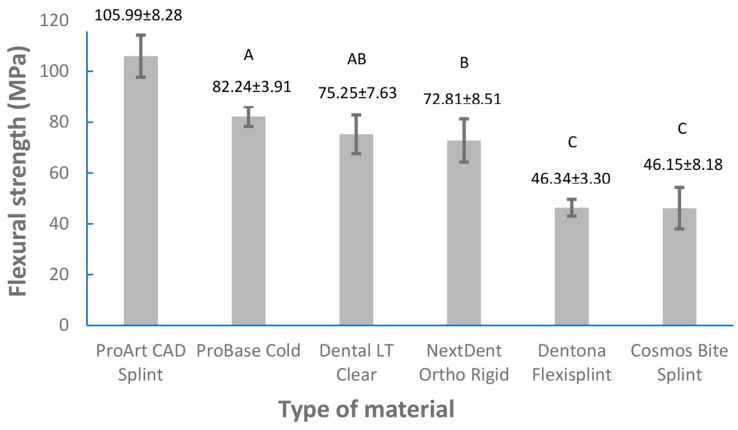
Mean values (±SD) of flexural strength for the tested groups. Identical uppercase letters above a column indicates that there is no significant difference between the tested groups (Tukey post-hoc test (*p* > 0.05)).

**Figure 3 dentistry-11-00199-f003:**
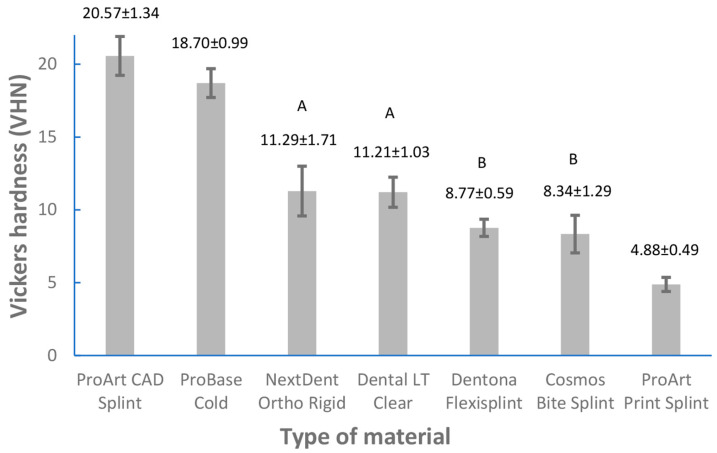
Mean values (±SD) of Vickers hardness for the tested groups. Identical uppercase letters above a column indicate that there is no significant difference between the tested groups (Tukey post-hoc test (*p* > 0.05)). VHN, Vickers hardness number.

**Table 1 dentistry-11-00199-t001:** Materials, composition, manufacturers, and occlusal splint fabrication techniques.

Material Name	Abbreviation	Composition	Manufacturer	Occlusal Splint Fabrication Method
NextDent Ortho Rigid	NOR	Monomer based on acrylic esters	Nextdent BV, Soesterberg, The Netherlands	3D printing
Dental LT Clear	DLC	Acrylic light-polymerizing resin	Formlabs, Somerville, MA, USA	3D printing
Dentona Flexisplint	DFS	Acrylic light-polymerizing resin	Dentona AG, Dortmund, Germany	3D printing
Cosmos Bite Splint	CBS	Acrylic light-polymerizing resin	Straumann, Basel, Switzerland	3D printing
ProArt Print Splint	PPS	Acrylic light-polymerizing resin	Ivoclar Vivadent AG, Schaan, Liechtenstein	3D printing
ProArt CAD Splint	PCS	PMMA	Ivoclar Vivadent AG, Schaan, Liechtenstein	Milling
ProBase Cold	PBC	PMMA	Ivoclar Vivadent AG, Schaan, Liechtenstein	Conventional cold polymerization

## Data Availability

All data are available in the manuscript.

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
