# Peer review of "Mechanical Properties of 3D-Printed Occlusal Splint Materials"

_dentistry, 2023, doi:10.3390/dj11080199_

Round 1

Reviewer 1 Report

Dear authors, thank you for submitting your very interesting article to The Journal of Dentistry. Certainly, your article is of significant clinical relevance, however some changes need to be done to the manuscript to ensure its final form is adequate for the readers of the journal. Please address the comments below.

ABSTRACT: The abstract is concise and well presented. However, please mention the results of the ANOVA and include the P values of the pairwise comparisons that presented statistical significance.

INTRODUCTION:

Besides reduction in material usage/wastage, and the possibility of manufacturing objects with complex geometries 3D printing presents several advantages over milling. Please elaborate more on this aspect.

Besides its use for the manufacture of occlusal splints, other appliances can be manufactured through additive manufacturing. Please mention other clinical applications of 3D printing in dentistry. Below are some articles as references demonstrating other clinically relevant uses of this manufacturing technology.

·         10.1016/j.prosdent.2022.04.012

·         10.1111/jopr.13556

MATERIALS AND METHODS:

Very throughout description of the materials and research methodology.

Please confirm if the AccuLab 3D printer by Shinning 3D is manufactured in Germany. As far as I know Shinning 3D headquarters are in Hangzhou China. Please confirm. Other than this the materials and methods are well written and clearly presented.

RESULTS:

Results are well presented. However, it would be good to have the significant pairwise differences with their respective P values mentioned in the results section.

DISCUSSION:

Double check your statement regarding the relationship between energy absorption and flexibility. Is the flexural strength important when it comes to resist high-impact forces? (Page 7 line 236)

Please add a few more recommendations for future research.

CONCLUSIONS:

Please double check your statement regarding the lack of statistically significant differences between DLC and PBC. Not statistically difference does not necessarily mean “the same”. Please reword your conclusions.

Author Response

Response to Reviewer 1:

Dear authors, thank you for submitting your very interesting article to The Journal of Dentistry. Certainly, your article is of significant clinical relevance, however some changes need to be done to the manuscript to ensure its final form is adequate for the readers of the journal. Please address the comments below.

1) The abstract is concise and well presented. However, please mention the results of the ANOVA and include the P values of the pairwise comparisons that presented statistical significance.

Response: The level of significance, results of the ANOVA test and P value are now included in the abstract.

2) Besides reduction in material usage/wastage, and the possibility of manufacturing objects with complex geometries 3D printing presents several advantages over milling. Please elaborate more on this aspect.

Response: Dear reviewer, advantages of 3D printing over milling are now listed in the introduction section of the manuscript, as recommended.

3) Besides its use for the manufacture of occlusal splints, other appliances can be manufactured through additive manufacturing. Please mention other clinical applications of 3D printing in dentistry. Below are some articles as references demonstrating other clinically relevant uses of this manufacturing technology.

  • 10.1016/j.prosdent.2022.04.012
  • 10.1111/jopr.13556

Response: Clinical applications of 3D printing are now added in the introduction section. Since 3D printing is used a lot in dentistry we focused on use of photopolymers in prosthodontics, in order not to lose the focus of the manuscript. Suggested articles are used to complement the manuscript.

4) Very throughout description of the materials and research methodology.

Please confirm if the AccuLab 3D printer by Shinning 3D is manufactured in Germany. As far as I know Shinning 3D headquarters are in Hangzhou China. Please confirm. Other than this the materials and methods are well written and clearly presented.

Response: Thank you! Headquarters of Shinning 3D is corrected to Hangzhou, China in the materials and methods section.

5) Results are well presented. However, it would be good to have the significant pairwise differences with their respective P values mentioned in the results section.

Response:  At the end of the results section, paragraph describing significant pairwise differences and P values is added, as recommended.

6) Double check your statement regarding the relationship between energy absorption and flexibility. Is the flexural strength important when it comes to resist high-impact forces? (Page 7 line 236)

Response: Dear reviewer, thank you for your concern. The statement is checked and left unmodified. Similar statement was previously published in the following study (Ucar Y, Akova T, Aysan I. Mechanical properties of polyamide versus different PMMA denture base materials. J Prosthodont. 2012;21:173-176.).

7)Please add a few more recommendations for future research.

Response: More recommendations for the future studies are added at the end of the discussion section.

8) Please double check your statement regarding the lack of statistically significant differences between DLC and PBC. Not statistically difference does not necessarily mean “the same”. Please reword your conclusions.

Response: Conclusions are reworded, as suggested.

Reviewer 2 Report

Thanks to the authors for outlining the conducted research and its results.

The introduction section of the manuscript is clear. However, it is recommended that the authors evaluate the wording they have chosen. A professional proofreading would enhance its quality.

Line 110 and 111: “3D printed materials for occlusal splint fabrication are in general inferior when compared to conventional PMMA or milled materials”. The authors should carefully check if they can confidently generalize that all 3D printed materials have inferior properties. That is keeping in mind that the mechanical properties of occlusal splints depend more on the material composition rather than the manufacturing technology.

In general it is a well written manuscript that requires some minor English editing/proofreading.

In general it is a well written manuscript that requires some minor English editing/proofreading.

Author Response

Response to Reviewer 2:

1) The introduction section of the manuscript is clear. However, it is recommended that the authors evaluate the wording they have chosen. A professional proofreading would enhance its quality.

Response: Thank you for the comment. English native speaker evaluated the manuscript, as recommended.

2) Line 110 and 111: “3D printed materials for occlusal splint fabrication are in general inferior when compared to conventional PMMA or milled materials”. The authors should carefully check if they can confidently generalize that all 3D printed materials have inferior properties. That is keeping in mind that the mechanical properties of occlusal splints depend more on the material composition rather than the manufacturing technology.

Response: Dear reviewer, thank you for the comment. We added one sentence after 110/111 line sentence, which points that mechanical properties of occlusal splints depend more on the material composition rather than the manufacturing technology.

3) In general it is a well written manuscript that requires some minor English editing/ proofreading.

Response: The manuscript is edited by English native speaker.

Reviewer 3 Report

Excellent study and a relevant topic to dental clinicians, manufacturers of 3D printing resins and the dental community as a whole. 

The Introduction and discussion are well written and sound. Data is well presented and easy to follow. A well written study. 

Minor editing at the bottom of page 3 to keep all the content in Table 1 together. 

Author Response

Response to Reviewer 3:

1) Excellent study and a relevant topic to dental clinicians, manufacturers of 3D printing resins and the dental community as a whole.

Response: Dear reviewer, thank you!

2) The Introduction and discussion are well written and sound. Data is well presented and easy to follow. A well written study.

Response: Thank you for the comment.

3) Minor editing at the bottom of page 3 to keep all the content in Table 1 together.

Response: Table 1 is edited and all the content is together now.

Reviewer 4 Report

The present manuscript titled as “Mechanical properties of 3D printed occlusal splint materials” reported the mechanical properties of 3D printed materials comparing with milled and conventional cold-polymerized occlusal splint materials and their results indicated that compared with milled and conventional cold-polymerized occlusal splint materials, although the flexural strength of 3D printed materials was not differed statistically from milled and conventional materials, the hardness (Vickers) values was lower in 3D printed occlusal splint materials. Over the manuscript was well organized and the data supports their results. 

Followings are some comments for the manuscript,

- Why the authors not accessing the wear behavior although the authors already mentioned about this substance in introduction with regards to the literature?

- Did the authors calculate the sample size? Please explain whether there was any power analysis done for the study.

- Line no 248 – pls identify the brand name?

Also, it would be more interesting for the readers/clinicians to point out the use of 3D printed occlusal splint at the short-term period rather than it would not recommend for long-term therapy

Author Response

Response to Reviewer 4:

1) The present manuscript titled as “Mechanical properties of 3D printed occlusal splint materials” reported the mechanical properties of 3D printed materials comparing with milled and conventional cold-polymerized occlusal splint materials and their results indicated that compared with milled and conventional cold-polymerized occlusal splint materials, although the flexural strength of 3D printed materials was not differed statistically from milled and conventional materials, the hardness (Vickers) values was lower in 3D printed occlusal splint materials. Over the manuscript was well organized and the data supports their results.

Response: Thank you!

2) Why the authors not accessing the wear behavior although the authors already mentioned about this substance in introduction with regards to the literature?

Response: Flexural strength and surface hardness are two properties most commonly used to define mechanical properties of a certain material (in this case occlusal splint material). In the literature there is a lack of data regarding flexural strength and surface hardness values of 3D printed materials. The opposite was seen regarding wear behavior, where more studies evaluated it. Recently, a systematic review was conducted on the suggested property (Grymak A, Aarts JM, Ma S, Waddell JN, Choi JJE. Wear Behavior of Occlusal Splint Materials Manufactured By Various Methods: A Systematic Review. J Prosthodont. 2022;31:472-487.), meaning that conclusions about wear behavior were already provided. Furthermore, wear behavior and surface hardness are two correlated properties, meaning that material with higher values of wear resistance has also higher values of surface hardness and vice versa.

3) Did the authors calculate the sample size? Please explain whether there was any power analysis done for the study.

Response: Power analysis was not performed for the present study. For this type of study, 10 specimens per group is considered normal. The same number of specimens (n=10) was used in similar previously published (contemporary) studies (Prpic V, Slacanin I, Schauperl Z, Catic A, Dulcic N, Cimic S. A study of the flexural strength and surface hardness of different materials and technologies for occlusal device fabrication. J Prosthet Dent. 2019;121:955-959.; Reymus M, Stawarczyk B. In vitro study on the influence of postpolymerization and aging on the Martens parameters of 3D-printed occlusal devices. J Prosthet Dent. 2021;125:817-823.; Wada J, Wada K, Garoushi S, Shinya A, Wakabayashi N, Iwamoto T, Vallittu PK, Lassila L. Effect of 3D printing system and post-curing atmosphere on micro- and nano-wear of additive-manufactured occlusal splint materials. J Mech Behav Biomed Mater. 2023;142:105799.).

4) - Line no 248 – pls identify the brand name?

Response: Response: Dear reviewer, at our document downloaded from Dentistry Journal website, there is no sentences at line 248. We have checked manuscript, and we added brand names at line 272 (line 286 in revised document with tracking changes on). Thank you for pointing it out.

5) Also, it would be more interesting for the readers/clinicians to point out the use of 3D printed occlusal splint at the short-term period rather than it would not recommend for long-term therapy.

Response: Thank you for this comment. We pointed out that 3D printed occlusal splints (materials) could be used for a short-term period therapy (at the end of the discussion section).

Reviewer 5 Report

Review Report on

Mechanical properties of 3D printed occlusal splint materials

Comments and Suggestions for Authors

The Aim of this study was to evaluate flexural strength and surface hardness of today’s 3D printed occlusal splint materials and compare it with milled and conventional cold-polymerized occlusal splint materials.

In the current context of increasing prevalence of digital technologies in all dental specialties, the study it is actual, even not extremely original.

The Manuscript it is clear, but some modification should be done:

1. The description of the methods used to obtain the samples is rather difficult to follow by the reader. I would recommend a separate paragraph for the three different techniques (3D Printing, Milling and conventional)

2. The presence of some images of the samples before testing and / or during the experiment into the Materials and Methods section, would be of interest and a plus for the manuscript

Author Response

Response to Reviewer 5:

1) The description of the methods used to obtain the samples is rather difficult to follow by the reader. I would recommend a separate paragraph for the three different techniques (3D Printing, Milling and conventional).

Response: Thank you for pointing this out. Separate paragraphs for different techniques (3D printing, milling and conventional) are added in the materials and methods section.

2) The presence of some images of the samples before testing and / or during the experiment into the Materials and Methods section, would be of interest and a plus for the manuscript.

Response: Dear reviewer, thank you for noticing. The scheme of specimens used in the present study is prepared and added into manuscript to help readers to understand the text more easily.
